# Comorbidity Patterns in Patients at Cardiovascular Hospital Admission

**DOI:** 10.3390/medicines10040026

**Published:** 2023-03-28

**Authors:** Cezara-Andreea Soysaler, Cătălina Liliana Andrei, Octavian Ceban, Crina-Julieta Sinescu

**Affiliations:** 1Department of Cardiology, University of Medicine and Pharmacy “Carol Davila”, Emergency Hospital “Bagdasar-Arseni”, 050474 Bucharest, Romania; 2Economic Cybernetics and Informatics Department, The Bucharest University of Economic Studies, 010374 Bucharest, Romania

**Keywords:** comorbidities, heart failure, cluster analysis, hospital admission

## Abstract

Hypertension frequently coexists with obesity, diabetes, hyperlipidemia, or metabolic syndrome, anditsassociation with cardiovascular disease is well established. The identification and management of these risk factors is an important part of overall patient management. In this paper, we find the most relevant patterns of hospitalized patients with cardiovascular diseases, consideringaspects of their comorbidities, such as triglycerides, cholesterol, diabetes, hypertension, and obesity. To find the most relevant patterns, several clusterizations were made, playing with the dimensions of comorbidity and the number of clusters. There are three main patient types who require hospitalization: 20% whose comorbidities are not so severe, 44% with quite severe comorbidities, and 36% with fairly good triglycerides, cholesterol, and diabetes but quite severe hypertension and obesity. The comorbidities, such as triglycerides, cholesterol, diabetes, hypertension, and obesity, were observed in different combinations in patients upon hospital admission.

## 1. Introduction

Despite a significant decrease in age-specific cardiovascular mortality over the past few decades, cardiovascular disease remains the primary cause of mortality, accounting for approximately 31% of deaths worldwide, as reported by the World Health Organization [1,2].

According to the guidelines provided by the American College of Cardiology/American Heart Association (ACC/AHA) and the European Society of Cardiology (ESC) for the diagnosis and management of chronic heart failure (CHF), heart failure (HF) is a clinical syndrome that results from a combination of various anatomical or physiological cardiac dysfunctions that impair the ventricles’ ability to be adequately supplied with and eject blood [3,4].

In developed nations, HF is found in approximately 1–2% of the adult population or occurs at a rate of 5–10 cases per 1000 people each year. [5,6].

Comorbidity refers to the presence of one or more chronic diseases alongside an index disease. In patients with heart failure (HF), comorbidities such as hypertension, diabetes mellitus, obesity, hyperlipidemia, and metabolic syndrome are prevalent and can impact clinical outcomes [7,8,9].

It is noteworthy that while hypertension, diabetes mellitus, obesity, hyperlipidemia, and metabolic syndrome are known to be associated with the development of heart failure (HF) in the general population, their impact on clinical outcomes and management of patients with established HF is uncertain and poses significant challenges.

Current guidelines have focused on the importance of lifestyle modifications, managing blood cholesterol levels [10,11], and addressing overweight and obesity [12] in both the general population and individuals with a heightened cardiovascular risk. Nevertheless, these guidelines do not explicitly outline how to manage such comorbidities in patients with heart failure (HF). 

Hypertension is a global epidemic. Within many countries, hypertension—defined as repeatedly elevated blood pressure (BP) exceeding 140/90 mm Hg—affects approximately 50% of individuals aged 60 and older. Despite the increased utilization of antihypertensive medications, the prevalence of hypertension is steadily rising [13]. It is commonly understood that hypertension is linked to increased cardiovascular and all-cause mortality, irrespective of other risk factors [14].

The significance of hypertension as a contributor to HF is well-established, particularly when left unmanaged. The remaining risk of hypertension for middle-aged and elderly individuals in the US is 90%, highlighting the enormous public health impact of this condition [15]. Therefore, it is imperative to implement effective strategies to control hypertension and prevent HF.

Various observational studies have indicated that individuals with diabetes mellitus have a higher likelihood of developing HF [16,17,18,19,20]. Additionally, studies have shown that even slight increases in glucose and abnormalities in insulin resistance—without the presence of overt diabetes mellitus—are associated with an increased risk of HF [21,22]. The contributing mechanisms that lead to a greater incidence of HF in people with diabetes mellitus are likely multifaceted.

Apart from these risk factors, diabetes mellitus can also lead to cardiac dysfunction independently, which is known as diabetic cardiomyopathy. Given the higher rates of morbidity and mortality associated with both diabetes mellitus and HF, it is becoming increasingly important to focus on prevention efforts. These efforts should include the treatment and prevention of coronary artery disease as well as adequate management of hypertension in individuals with diabetes mellitus.

Numerous studies have demonstrated the association between obesity and the risk of developing heart failure (HF). While the notion of cardiomyopathy related to obesity has been previously described [23,24,25], the robust and independent relationship between obesity, as measured by body mass index (BMI), and the incidence of HF has only recently been established.

A study conducted on 5,881 participants of the Framingham Heart Study demonstrated that for every 1-unit increase in BMI, there was a 5% increase in the risk of HF for men and a 7% increase for women. This increase in risk was found to be independent of demographics and known risk factors, including MI, diabetes mellitus, hypertension, and cholesterol [26].

While obesity is widely recognized as a risk factor for cardiovascular diseases (CVD) and the development of heart failure (HF), it is interesting to note that obesity, as determined by BMI or other anthropometric measures, does not appear to be a risk factor for negative outcomes in patients who already have established HF. This phenomenon, known as the obesity paradox, has been extensively studied in the medical literature related to HF. Considering the significant prevalence of obesity in both the general population and among those with HF, it is important to have discussions about the treatment of obesity.

Research has consistently demonstrated that hypercholesterolemia is linked to adverse outcomes such as mortality, cardiovascular events, and the development of heart failure (HF) in both the general population and among patients with atherosclerotic cardiovascular diseases (CVD) [27,28]. These conditions are quite common and are known to contribute to the development of HF in the general population

To sum up, implementing lifestyle changes and managing comorbidities continue to pose significant challenges for patients with heart failure (HF). Despite the implementation of intensive lifestyle modifications and medical interventions, preventing and effectively treating hypertension, diabetes mellitus, and obesity have proven to be challenging in patients with HF.

It is important to note that even small changes made in the prevention of comorbidities and risk factors for heart failure (HF) can have a significant impact on the development and outcomes of HF. This highlights the crucial role that prevention strategies play in reducing the overall burden of HF, as they are likely the most effective approach in doing so.

## 2. Materials and Methods

### 2.1. Methodology

Since we need a way to better understand what kind of patients with cardiovascular disease symptoms show up for hospitalizationregardingcomorbiditydimensions, unsupervised machine learning techniques for clustering were used. We used k-means clustering with several clusters, and for each number of clusters, we measuredthesilhouette score and sum of squares within. The distance used is the classical Euclidian. We had a total of 199 patients hospitalized for cardiovascular disease symptoms in the Hospital Cardiology Clinic of the emergency “BagdasarArseni”inBucharest. The clustering process involved dividing the patients into groups so that the patients in one group are as similar as possible to each other and less similar to those in other groups. This way, the main groups of patients in a sample could be seen, and assumptions could be made about the entire population. The data were processed using python programming language, and the *scikit-learn* package was used for clustering.

### 2.2. Data

Comorbidity dimensions:TG_high: binary variable with 1 when triglycerides are over 200, else 0;COLES_high: binary variable with 1 when cholesterol is over 240, else 0;Obesity: binary variable with 1 when the patient has any stages of obesity (BMI over 30), else 0;Has_diabetes: binary variable with 1 when patient has diabetes, else 0;HTA_intensity: ordinal variable for hypertension. This reflects the hypertension intensity and is made based on hypertension stages; no hypertension has a value of 0, stage 1 has a value of 0.33, stage 2 has a value of 0.66, and stage 3 has a value of 1.

Clustering using variables in different scales is challenging. The use of binary variables for triglycerides and cholesterol was preferred to be on the same scale as the other variables. There was also the option of normalization with values between 0 and 1, although, from a medical perspective, it makes more sense to show whether a patient has the value after a certain threshold for these variables. For the diabetes variable, the patients with diabetes were in different stages: some were only following the diet (26%), some were taking antihyperglycemic drugs treatment (48%), and the rest were on insulin treatment (26%). Regarding the variable hypertension intensity, the range from 0 to 1 was divided equally according to the stages. Thus, the distance between a patient without hypertension and a patient with hypertension stage 1 was the same as the distance between a patient with stage 2 and a patient with stage 3. This method was preferred considering the number of patients in the sample and the ease of interpreting the results.

### 2.3. Descriptive Statistics

#### Comorbidity Dimensions

The analyzed patients are between 40 and 93 years old. Half are between 63 and 79 years old. In the sample of patients, 53% are women and 47% are men. We did not introduce these variables as clustering dimensions because we wanted to consider only comorbidities. Figure 1 shows the proportion of comorbidities in patients, and Figure 2 shows hypertension intensity. The blue bars, with 1, reflect the percentage of patients having that comorbidity, and the orange bars, with 0, reflect the percentage of patients who don’t.

There are 50% diabetics in the sample. Regarding triglycerides, only 8% have values above 200, and only 13% have cholesterol values above 240. Overall, 79% have a BMI above 30.

Regarding hypertension at admission, 50% were in a serious condition. A total of 36% were in stage 2, and 7% were in stage 1, as were those without hypertension.

There was no very high correlation between comorbidities. There was a slight trend of correlation between high cholesterol and high triglycerides.

## 3. Results

The main goal after this exploratory analysis was to understand better the patterns regarding the comorbidities of the hospitalized patients. This could be achieved using classic clustering algorithms. The patients were divided into clusters, the patterns found were reflected, and several metrics were calculated. These helpedselect the number of clusters, but they only had an advisory role. Other characteristics were also important, such as the ease of interpreting the results and the number of patients available because, with a large number of clusters, there may be very few patients in them and they would no longer be so representative. Figure 3 shows the sum of squares within the metric for each number of clusters.

In Figure 4a it can be seen that as k increases, the dissimilarity in the clusters starts to decrease, and the patients become more similar. For example, from k equal 4 to k equal 5, you do not gain much divided variance, as this is a decrease of only 19% compared to 28% from k equal 2 to k equal 3. To have another criterion for choosing k, the silhouette metric is also analyzed, which can be between −1 and 1, where 1 shows a perfect separation between clusters. This is based on the distances between the objects to other objects in the same cluster and, respectively, to objects in the other clusters.

We notice in Figure 4b thatas k increases the silhouette score increases, but the complexity of the interpretation also increases. As k increases, the number of observations in the clusters will decrease, and the risk of not representing a cluster very well increases. In other words, if a cluster is not well represented, it might not actually exist at the population level.

Figure 5 shows the evolution of the number of observations from the cluster with the fewest observations for each k. So, if we split the patients into two clusters, the cluster with the fewest observations has 99; if we split them into three clusters, the cluster with the fewest has 40, and so on. It is relevant to measure this because, for each k, we have an idea concerning the risk of having too few observations in a cluster. It can be seen that with k equal to 3, we have 40 patients in the smallest cluster, and in the analysis with 4 clusters only 28 patients remain. The sum of squares within and silhouette metrics would suggest that 4 clusters would have more similar observations and, thus, a better representation of patterns than 3 clusters. There is not, however, a very big difference between the performance of these two scenarios, with 3 and 4 clusters, respectively, and in the case of 5 clusters, performance would remain similar. The number of observations for 3 clusters is much better than with 4 clusters, where there is a greater risk of not representing those clusters well. Considering these and the ease of interpreting 3 clusters instead of 4, we choose to divide the patients into 3 clusters.

The main patterns are defined by patients with the following characteristics-Figure 6:

Cluster 1:

The first type of patients at admission presentthemselvesbest compared to other clusters, taking hypertension into account, with an average intensity of 0.63. Half of the patients have stage 2 hypertension, a quarter have stage 3, and the other quarter have stage 1 or no hypertension. Only 30% have diabetes. Only 5% have high triglycerides, the same for cholesterol. None of them have obesity. Compared to the other groups, these patients seem the most stable. Out of the total number of patients, this type of patient represents 20%.

Cluster 2:

These patients have the worst characteristics. They all have diabetes, a quarter have high cholesterol, and 11% have high triglycerides. Most of them have stage 3 hypertension, and all are obese. This type of patient represents 44%.

Cluster 3:

This type of patient hasvarious types of hypertension. Half have stage 3, 20% have stage 1 or none, and the remainder have stage 2. Of these, none have diabetes, all have obesity, 6% have high triglycerides, and 7% have high cholesterol. They represent 36% of patients.

The graph and table above represent the analysis with 3 clusters. In the annex, you can also find the characteristics for 2, 4, and 5 clusters.

To better understand the patients and how they are divided into the 3 clusters, the following image was created. The patients were represented in 2d, starting from the dimensions related to comorbidities, and the first two principal components were extracted using Principal Component Analysis-Figure 7.

These components extract 55% of the original information from the 5 dimensions related to comorbidities. Each point represents a patient, although by using 4 binary variables and an ordinal variable with 4 possible values, many points overlap. The closer two points are, the more similar they are. This image depicts patients with their comorbidities at admission. In the factor matrix below (Figure 8), some interpretations are suggested for main components 1 and 2.

The factor matrix shows the Pearson correlation of the new components with the original variables. These two extract the maximum information from the original variables. It can be seen how PC1 is moderately correlated with all variables. Thus, we can interpret that the higher PC1 is, the worse a patient is. Regarding PC2, there is a strong correlation with high triglyceride and a slight correlation with high cholesterol, but also a slight inverse correlation with hypertension intensity and obesity. With higher PC2, the patient tends to have high triglycerides and high cholesterol but also lower hypertension and obesity, although there are smaller correlations.

Returning to the scatterplot graph, the further a patient is to the right, the more comorbidities they tend to have. It can be seen that they are predominantly from cluster 2. Those from clusters 1 and 2 have certain similarities, especially concerning hypertension, but they are distinguished by obesity and diabetes.

## 4. Discussion

In the current study, we wanted to find the most relevant patterns of hospitalized patients taking into account some dimensions of their comorbidities, triglycerides, cholesterol, diabetes, hypertension, and obesity. For this, several clusterizations playing with dimensions of comorbidity and the number of clusters were made.

After analyzing the existing data, we established three main patterns of patients who showed up for hospitalization: Cluster 3 (36%)had pretty good triglycerides, cholesterol, and diabetes but quite severe hypertension and obesity;Cluster 2 (44%)had severe comorbidities; Cluster 1 was the most stable group, with 20% of patients having not-so-severe comorbidities.

Atherogenic dyslipidaemia, hyperglycemia, visceral obesity, and hypertension tend to occur in clusters in the same individuals. The high prevalence of dyslipidaemia reported in this study is consistent with this expectation.

Hypertension, diabetes mellitus, obesity, and hyperlipidemia are common in patients with heart failure (HF) and affect clinical outcomes. Although these comorbidities are associated with the development of HF in the general population and in patients with established HF, their contributory roles to clinical outcomes are not predictable, and their management is quite challenging.

These findings indicate that health education should be promoted among patients to improve dyslipidaemia, diabetes, hypertension, and obesity. Overall, the findings of this study show that the prevention and treatment of dyslipidaemia, diabetes, hypertension, and obesity should be prioritized in the at-risk population.

The main limitations of the study relate to the sample. We had relatively few patients from a single location. Thus, the obtained patterns could differ in other areas or even over time. For better estimates of how these comorbidities are found in patients admitted with heart disease, more studies should be done on more patients from more hospitals.

Recent guidelines have addressed the role of lifestyle modification, treatment of blood cholesterol, and management of overweight and obesityin the general population and in patients with increased cardiovascular risk. These guidelines, however, do not specifically address the management of such comorbidities in patients with HF. Improvements in primary prevention and pharmacological treatment of cardiovascular risk factors result in a lower risk of cardiovascular events and fewer deaths [29,30].

As a result, the age at which people have their first cardiovascular event would shift to a later age, andpatients would survive longer after the onset of cardiovascular disease [31]. At present, more patients with cardiovascular disease have a longer life expectancy and increasingly face comorbid conditions, making this population particularly interesting when studying comorbidity.

Comorbidity could lead to poorer functional status, lower quality of life, and even increased mortality [32,33]. In addition to the effects on patients, having multiple chronic diseases is also challenging for healthcare systems, such as primary care, because, traditionally, these are configured around single diseases.

In addition to a better understanding of the avatar of a hospitalized patient with cardiac problems, this study can also help with patient management in similar hospitals with cardiological hospitalizations. Knowing the main comorbidities and their proportion in patients, both in those analyzed here and in others to be analyzed, efficiency in hospitals could increase by forming teams of doctors specialized in comorbidities, allocating hospital resources, such as equipment, more efficiently.

## Figures and Tables

**Figure 1 medicines-10-00026-f001:**
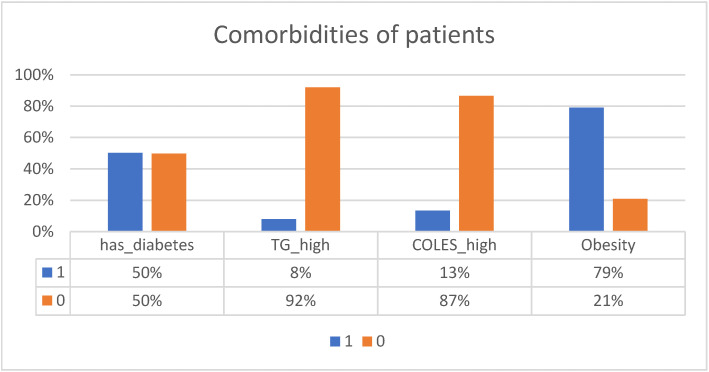
Comorbidities of patients.

**Figure 2 medicines-10-00026-f002:**
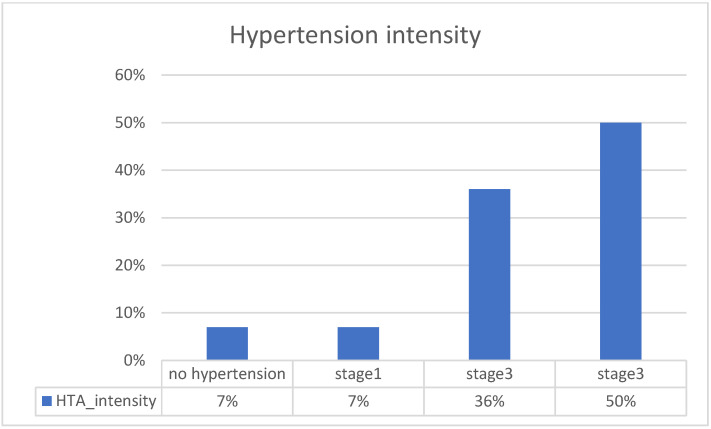
Hypertension intensity.

**Figure 3 medicines-10-00026-f003:**
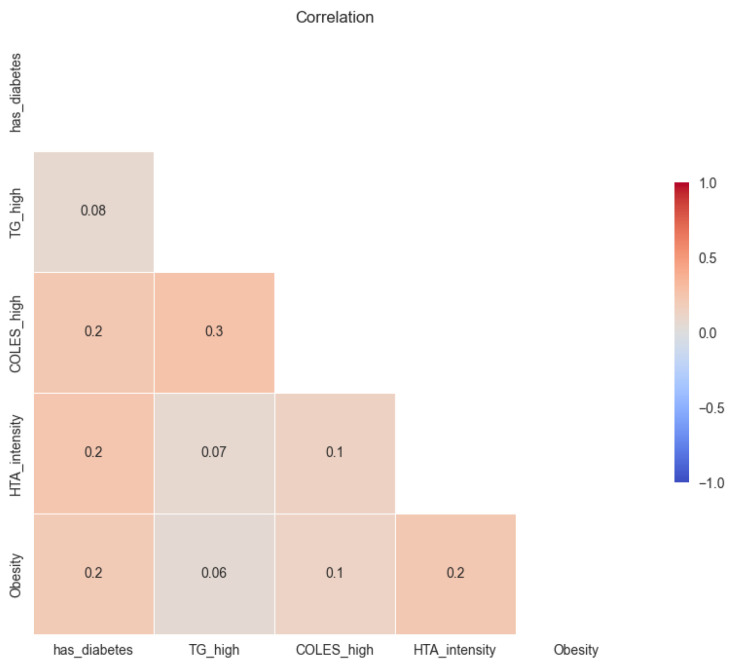
Correlation of comorbidities variables.

**Figure 4 medicines-10-00026-f004:**
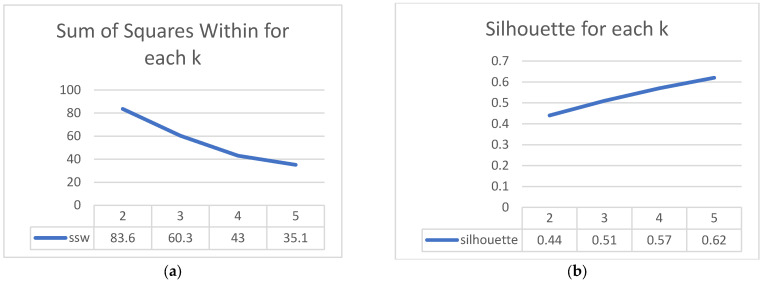
Clustering performance visualization. (**a**) Sum of squares within the plot showing how similar objects are in a cluster; (**b**) Silhouette score plot shows the quality of clusters.

**Figure 5 medicines-10-00026-f005:**
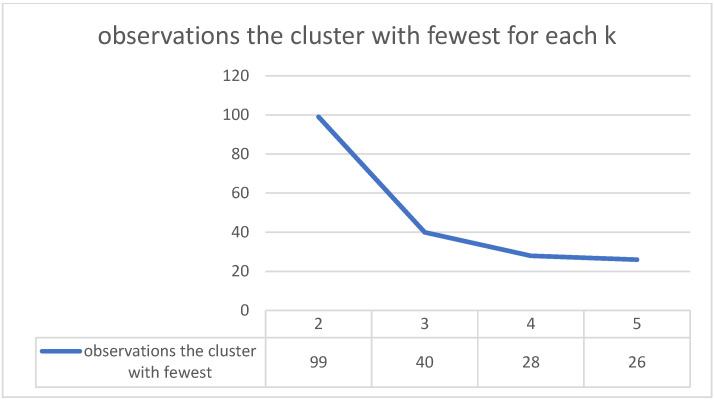
Number of observations for each cluster.

**Figure 6 medicines-10-00026-f006:**
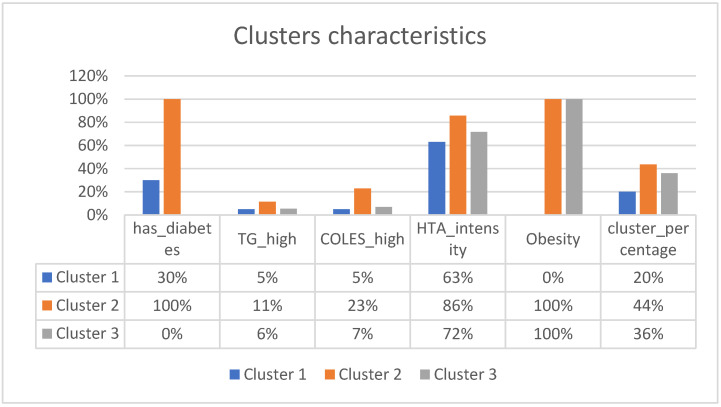
Clusters characteristics.

**Figure 7 medicines-10-00026-f007:**
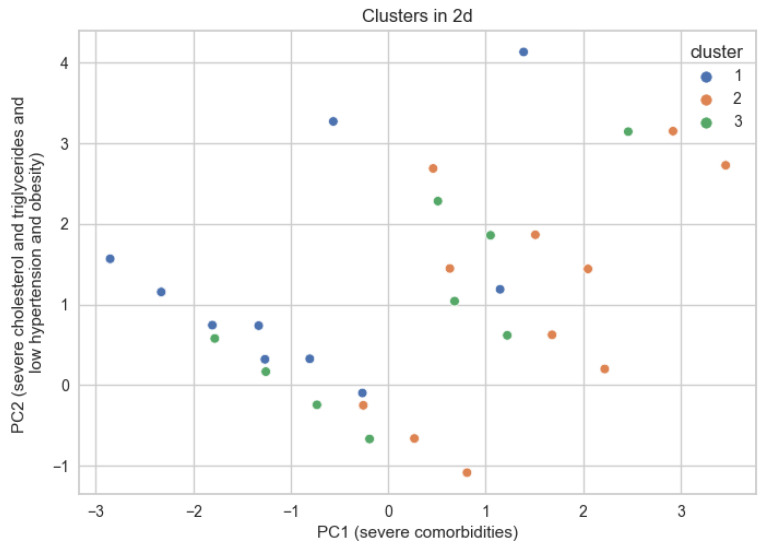
Observations grouped by cluster in 2d space.

**Figure 8 medicines-10-00026-f008:**
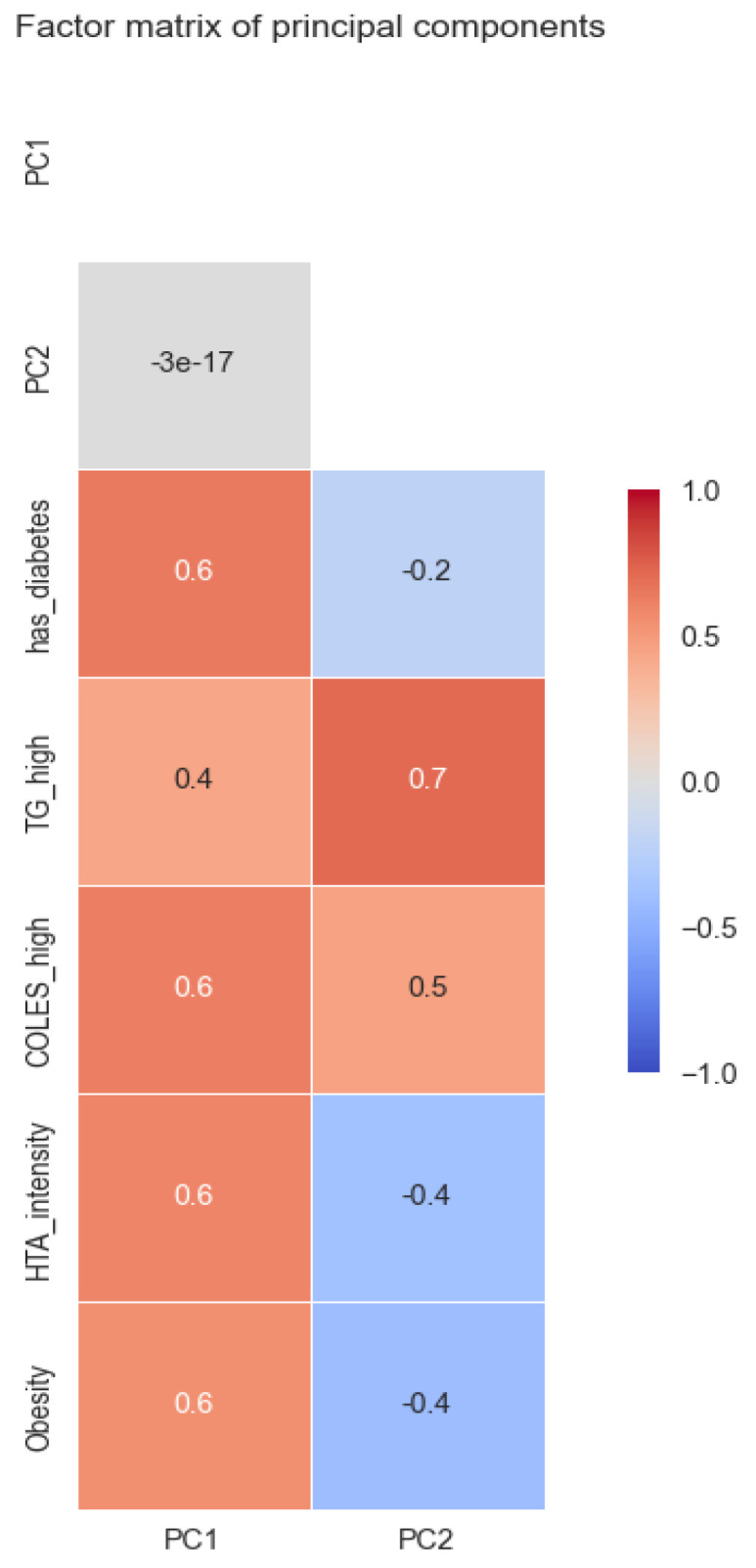
Factor matrix of the principal components.

## Data Availability

Data is available on request due to ethical restrictions. The data presented in this study are available on request from the corresponding author. The data are not publicly available due to ethical restrictions.

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
