# Peer review of "Comorbidity Patterns in Patients at Cardiovascular Hospital Admission"

_medicines, 2023, doi:10.3390/medicines10040026_

Round 1

Reviewer 1 Report

The Authors presented a single-center retrospective observational study on a small patient sample allegedly aiming to describe and classify into clusters the patients referred to their Hospital Cardiology Clinic for Heart Failure based on comorbidities. Comorbidities taken into account were: triglycerides and cholesterol levels, hypertension, obesity and diabetes. As far as this reviewer understands, machine learning methods were used to define clusters of patients: what was deemed by the Authors to be the best clustering method was one identifying three comorbidity patterns: - no severe comorbidities (20%); - severe comorbidities (44%); - severe hypertension and obesity (36%). This reviewer could not find in any part of the study a declaration of the starting hypothesis as well as a conclusion on practical relevance of the results.

This reviewer sees a number of issues with this manuscript:

- the introduction is too long and in the form of a degree thesis rather than scientific paper

- the comorbidities analyzed were chosen arbitrarily

- the reader would expect that after the identification of clusters some correlation with clinical outcomes would have been studied, but there is none

- the conclusions are vague, generical and inconsistent

- English language is poor, making reading the manuscript very difficult

Author Response

“This reviewer could not find in any part of the study a declaration of the starting hypothesis as well as a conclusion on practical relevance of the results. “

We didn’t start with some hypothesis to test, it was just an exploratory analysis (where clustering is the most classical type of exploratory analysis) where the results are some patterns found. The following paragraph it was added in idea to reflect practical relevance of the results:

In addition to a better understanding of the avatar of a hospitalized patient with cardiac problems, this study can also help the management of patients in other similar hospitals with cardiology hospitalizations. Knowing the main comorbidities, their proportion in patients, both those analyzed here and others to be analyzed, efficiency in hospitals could increase by forming teams of doctors specialized in comorbidities, allocating hospital resources, such as equipment, more efficiently.

- the introduction is too long and in the form of a degree thesis rather than scientific paper

In introduction we have pointed out aspects like the importance of comorbidities in cardiology diseases and how may relate some of them with the evolution of a disease or patients healthcare.

- the comorbidities analyzed were chosen arbitrarily

We chose them from several dimensions of patients available in the collected database. We had dimensions like social variables (education, income, region etc.), laboratory results, condition of the patients at admission etc. Those variables were not really arbitrarily selected, we were focus on comorbidities, and those selected comorbidities, taken together, described some relevant patterns (more or less). We have ran a lot of simulations and experiments with others variables and we didn’t want to complicate too much the article, since we didn’t have a large sample of patients we were focused only on a few, but most relevant, comorbidities in order to create as good as possible representative clusters.

- the reader would expect that after the identification of clusters some correlation with clinical outcomes would have been studied, but there is none

This would have been really interesting to see, we are in the process of collecting more data because we would like to really test some hypothesis here, and since having a small sample we would have a low statistical power. Actually, it was a trade-off between the size of patients in clusters (we could have control it by playing with the comorbidities included) which would help to further correlations studies, and the adding more comorbidities to get an interesting profile. We are aware of that, thank you for pointing this out, we are taking into consideration for further articles, but for that it would help more clear defined clusters, maybe having a suggestive name, for this studies this is what we have got considering the data.

- the conclusions are vague, generical and inconsistent

We added a new paragraph of conclusions

- English language is poor, making reading the manuscript very difficult

We are going to check it again and rephrase where it may be confusing

Reviewer 2 Report

The current manuscript concerns the study of the comorbidities found in cardiovascular patients in a specific hospital. It is an interesting study, however, some modifications should be done:

- Figure 1 data should say in the caption what the results are being represented by: is it mean? Median? And what about error bars, why are they not present?; same with figure 2;

- Figure captions should describe the meaning of used abbreviations;

- Figure 3 and figure 7 should be bigger, in order to provide better reading and comprehension;

- Data analysis should include the P value, and level of confidence;

- If “Cluster 1”, “Cluster 2”, “Cluster 3” are meant to be subsections, they should be numbered;

- A paragraph should be added in the discussion section mentioning the limitations of the current study, and how the conclusions only truly apply to the studied sample, which was of only 1 hospital, in 1 location;

- A conclusion section is missing after the “4. Discussion” section, with the main conclusions of the study, and future perspectives.

Author Response

- Figure 1 data should say in the caption what the results are being represented by: is it mean? Median? And what about error bars, why are they not present?; same with figure 2;

Thank you for your observation, I added some explanatory text for figure 1, an for figure 2 it is explained on the second paragraph after it, line 158.

- Figure captions should describe the meaning of used abbreviations;

We didn’t want to put redundant text. We thought that figure 1 has the variables described before, starting with line 121, they are the main features used; same for figure 2,3,5; for figure 3 and 4 there is some text under it; figures 6,7 discuss about Principal Component Analysis and is quite suggestive where PC came from

- Figure 3 and figure 7 should be bigger, in order to provide better reading and comprehension;

We’ve updated them, thank you!

- Data analysis should include the P value, and level of confidence;

In data analysis we just counted the patients having comorbidities, we don’t have tests here or estimations of parameters.

- If “Cluster 1”, “Cluster 2”, “Cluster 3” are meant to be subsections, they should be numbered;

We thought they are part of the “Results” sections so we didn’t want to complicate the structure having subchapters

- A paragraph should be added in the discussion section mentioning the limitations of the current study, and how the conclusions only truly apply to the studied sample, which was of only 1 hospital, in 1 location;

Done, starting with line 273

- A conclusion section is missing after the “4. Discussion” section, with the main conclusions of the study, and future perspectives.

We added some paragraph related to that, line 298.

Reviewer 3 Report

Major Comments:

1)      This article does not read particularly easily. Given the complexities of the analysis conducted it could benefit significantly in terms of reader ease from English editing (Introduction, methods and results section).

2)      Line 89-93: Authors discuss hypercholesterolemia as being associated with worse outcomes but then go on to whether low cholesterol levels play a causative role etc. Please clarify if you are discussing hyper or hypo, alternatively if you are discussing both the paragraph requires restructuring and additional clarifications.

“In the general population and in patients with atherosclerotic CVD, hypercholesterolemia has consistently been shown to be associated with worse outcomes, including mortality, cardiovascular events, and the development of HF [27,28]. It is currently unclear whether low cholesterol levels play a causative role in the worse outcome of patients with HF or whether low cholesterol levels merely reflect an advanced disease state.”

3)      The introduction appears to focus mainly on heat failure as the primary discussion point while the actual analysis appear to evaluate  those hospitalised for CVD as a whole and does not appear to relate directly to HF. Please clarify if these hospitalisations are only for those in HF, if not CVD as a whole should be better integrated into the introduction.

4)      Please discuss the strengths and limitations of this type of model.

Minor Comments:

1)      Lines 43-46. Sentences is repeated:” …and in patients with increased cardiovascular risk, but, these guidelines did not specifically address the management of such comorbidities in patients with HF. However, these guidelines did not specifically address the management of such comorbidities in patients with HF.”

2)      Line 70-77 Sentence is repeated: “Multiple studies have established obesity as a risk factor for the development of HF. Although the concept of cardiomyopathy relating to obesity has been described previously [23-25] the strong, independent, incremental relationship between obesity as indexed by body mass index (BMI) and HF incidence was only more recently established. Multiple studies have established obesity as a risk factor for the development of HF. Although the concept of cardiomyopathy relating to obesity has been described previously, [23-25] the strong, independent, incremental relationship between obesity as indexed by body mass index (BMI) and HF incidence was only more recently established.”

3)      Line 102: “HF.Theseunderline” Spaces are missing.

Author Response

Major Comments:

1.This article does not read particularly easily. Given the complexities of the analysis conducted it could benefit significantly in terms of reader ease from English editing (Introduction, methods and results section).

We are going to check again and rephrase where the text might be confusing

  1. Line 89-93: Authors discuss hypercholesterolemia as being associated with worse outcomes but then go on to whether low cholesterol levels play a causative role etc. Please clarify if you are discussing hyper or hypo, alternatively if you are discussing both the paragraph requires restructuring and additional clarifications.

“In the general population and in patients with atherosclerotic CVD, hypercholesterolemia has consistently been shown to be associated with worse outcomes, including mortality, cardiovascular events, and the development of HF [27,28]. It is currently unclear whether low cholesterol levels play a causative role in the worse outcome of patients with HF or whether low cholesterol levels merely reflect an advanced disease state.”

We are going to check again and rephrase where the text might be confusing

3.The introduction appears to focus mainly on heat failure as the primary discussion point while the actual analysis appear to evaluate  those hospitalised for CVD as a whole and does not appear to relate directly to HF. Please clarify if these hospitalisations are only for those in HF, if not CVD as a whole should be better integrated into the introduction.

The patients included in this study are diagnosed with ischemic heart disease and at admission show signs and symptoms of heart failure.

4.Please discuss the strengths and limitations of this type of model.

Thank you for the feedback, we’ve added a paragraph discussing about that, specially about limitations, startling with line 273.

Minor Comments:

  • Lines 43-46. Sentences is repeated:” …and in patients with increased cardiovascular risk, but, these guidelines did not specifically address the management of such comorbidities in patients with HF. However, these guidelines did not specifically address the management of such comorbidities in patients with HF.”

Solved, thank you!

  • Line 70-77 Sentence is repeated: “Multiple studies have established obesity as a risk factor for the development of HF. Although the concept of cardiomyopathy relating to obesity has been described previously [23-25] the strong, independent, incremental relationship between obesity as indexed by body mass index (BMI) and HF incidence was only more recently established. Multiple studies have established obesity as a risk factor for the development of HF. Although the concept of cardiomyopathy relating to obesity has been described previously, [23-25] the strong, independent, incremental relationship between obesity as indexed by body mass index (BMI) and HF incidence was only more recently established.”

Solved, thank you!

3)      Line 102: “HF.Theseunderline” Spaces are missing.

Solved, thank you!

Round 2

Reviewer 1 Report

I have no other requests.

Reviewer 3 Report

I am satisfied with the author response and changes.